# Cytological Samples: An Asset for the Diagnosis and Therapeutic Management of Patients with Lung Cancer

**DOI:** 10.3390/cells12050754

**Published:** 2023-02-27

**Authors:** Diane Frankel, Isabelle Nanni, L’Houcine Ouafik, Laurent Greillier, Hervé Dutau, Philippe Astoul, Laurent Daniel, Elise Kaspi, Patrice Roll

**Affiliations:** 1Aix Marseille University, APHM, INSERM, MMG, la Timone Hospital, Cell Biology Laboratory, 13005 Marseille, France; 2Aix Marseille University, APHM, CHU Nord, Oncobiology Laboratory, 13015 Marseille, France; 3Aix Marseille University, CNRS, INP, Inst Neurophysiopathol, 13005 Marseille, France; 4Multidisciplinary Oncology and Therapeutic Innovations, Marseille University Hospital, APHM, 13015 Marseille, France; 5Department of Thoracic Oncology, Pleural Diseases and Interventional Pulmonology, APHM, 13015 Marseille, France; 6Aix Marseille University, APHM, Anatomopathology Department, 13005 Marseille, France

**Keywords:** cytological sample, cytopathology, lung cancer, adenocarcinoma, PD-L1, molecular testing, NGS, immunocytochemistry

## Abstract

Background: Lung cancer has become the leading cause of cancer death for men and women. Most patients are diagnosed at an advanced stage when surgery is no longer a therapeutic option. At this stage, cytological samples are often the less invasive source for diagnosis and the determination of predictive markers. We assessed the ability of cytological samples to perform diagnosis, and to establish molecular profile and PD-L1 expression, which are essential for the therapeutic management of patients. Methods: We included 259 cytological samples with suspected tumor cells and assessed the ability to confirm the type of malignancy by immunocytochemistry. We summarized results of molecular testing by next generation sequencing (NGS) and PD-L1 expression from these samples. Finally, we analyzed the impact of these results in the patient management. Results: Among the 259 cytological samples, 189 concerned lung cancers. Of these, immunocytochemistry confirmed the diagnosis in 95%. Molecular testing by NGS was obtained in 93% of lung adenocarcinomas and non-small cell lung cancer. PD-L1 results were obtained in 75% of patients tested. The results obtained with cytological samples led to a therapeutic decision in 87% of patients. Conclusion: Cytological samples are obtained by minimally invasive procedures and can provide enough material for the diagnosis and therapeutic management in lung cancer patients.

## 1. Introduction

Lung cancer has become the leading cause of cancer death for men and women [1]. The management of this cancer has evolved over the last decade with the emergence of new therapies, such as tyrosine kinase inhibitors and immunotherapy. Therefore, patient samples must allow for both the diagnosis and molecular testing, as well as PD-L1 quantification. As patients are often diagnosed at an advanced stage [2], pathologists must use samples carefully and appropriately, as the diagnosis is no longer the only result needed for patient management. Cytological samples have a place in the management of patients with pulmonary mass, especially those with an advanced disease for whom surgery is not a therapeutic option. These patients with advanced diseases account for 45% of patients diagnosed with lung cancer [2]. The last version of the World Health Organization (WHO) classification granted a section entirely dedicated to cytology in lung cancer, showing the importance of these samples in this context [3].

In this article, we report the ability of cytological samples to perform the diagnosis of lung cancer and to obtain critical results, such as molecular profile and PD-L1 expression, which are essential for the therapeutic management.

## 2. Materials and Methods

### 2.1. Samples Collection

This study included cytological samples in which suspicious cells were observed and where immunocytochemistry was performed to characterize these cells. The samples were collected between January 2021 and September 2022 in the Cell Biology Laboratory (Timone Hospital, Assistance Publique des Hôpitaux de Marseille, Marseille, France). 

The different types of these cytological samples were pleural, pericardial, and peritoneal effusions, bronchoalveolar lavage fluids, endobronchial ultrasound guided transbronchial needle aspiration (EBUS-TBNA) lymph nodes, EBUS-TBNA mediastinal or pulmonary mass, cerebrospinal fluid, and bone marrow aspiration. Samples were not initially fixed and were kept at 4 °C until slide preparation (smears or cytospin). Slides were stained with Papanicolaou and May-Grünwald–Giemsa stains.

The conventional cytological diagnosis was performed by the Cell Biology Laboratory. PD-L1 testing was performed by the Anatomopathology Laboratory (Assistance Publique des Hôpitaux de Marseille, Marseille, France). The next generation sequencing (NGS) was performed by the Oncobiology Laboratory (Assistance Publique des Hôpitaux de Marseille, France). Samples included in this study were obtained from patients attending the Assistance Publique des Hôpitaux de Marseille for diagnosis and treatment. Results of the molecular testing and clinical data were retrospectively analyzed. This project was approved by the local ethics committee (PADS22-389).

### 2.2. Immunocytochemistry on Cytospins to Phenotype Tumor Cells 

Samples were prepared on cytospins as previously described in [4], following the manufacturer’s instructions. As a minimum, one wash was performed between each step. Slides were fixed with paraformaldehyde (PAF) 4% for 10 min, and then incubated with the peroxidase-blocking solution for 30 min. After being washed, slides were incubated with SensiTEK HRP kit (ScyTek Laboratories, Logan, UT, USA) for 10 min. Primary antibodies were incubated for 30 min (see Appendix A for the list of primary antibodies). Then, the biotinylated secondary antibody was incubated for 15 min, followed by Streptavidin/HRP for 20 min and DAB Quanto chromogen (Diagomics, Blagnac, France) for 5 min. Nuclei were counterstained with Mayer’s hemalun solution. Slides were mounted with Aquatex^®^ (Merck Millipore, Darmstadt, Germany). Slides were observed under optical microscope (Leica, Wetzlar, Germany). Mouse isotype IgG and rabbit polyclonal antibodies were used as negative controls as part of best practice method.

### 2.3. Immunohistochemistry on Cytoblock for PD-L1 Expression 

Cytoblocks were prepared to perform PD-L1 testing. Cytological samples were fixed with formalin 4% for 6 h then centrifugated for 5 min at 670 g. The supernatant was discarded. The cytoblock^TM^ kit (Epredia, Kalamazoo, MI, USA) was used to prepare cytoblocks following the manufacturer’s instructions. A slide stained with H&E was systematically performed before PD-L1 immunostaining to confirm the cytoblock quality and evaluate the adequate number of tumor cells.

PD-L1 immunostaining (QR001, Quartett, Germany) was performed with the optiview DAB detection Kit on Benchmarck Ultra (Ventana, Roche, Bale, Switzerland). A positive control was systematically performed as part of best practice method. 

### 2.4. Next Generation Sequencing

NGS was performed from frozen cell pellets as previously described [5]. In short, total nucleic acids were extracted with the Maxwell RSC Cell DNA Kit (Promega, Madison, WI, USA) and RNAs were extracted with the Maxwell RSC Simply RNA Blood Kit (Promega). The detection of mutations and fusions were performed by NGS on the Ion Torrent S5XL (ThermoFisher, Waltham, MA, USA) with a custom panel Oncomine Solid Tumor and Oncomine Solid Tumor+ (OST/OST+) and Oncomine Focus RNA assay kit (ThermoFisher, Waltham, MA, USA) (see Appendix A for the fusion transcript panel and the mutation transcript panel). Ion Torrent Suite, Ion Reporter software (ThermoFisher, Waltham, MA, USA) and a pipeline developed in our laboratory were used for the interpretation of the results.

## 3. Results

### 3.1. General Results 

Between January 2021 and September 2022, 259 cytological samples containing cells suspected of malignancy were analyzed by immunocytochemistry to characterize the type and origin of cancer.

Immunocytochemistry allowed for the characterization of the type of cancer in 248 cases (95.7%). In 59 samples (mostly pleural and peritoneal effusions), the immunocytochemistry confirmed the malignancy but with another origin than lung (for example, ovarian, breast, colorectal or pancreatic carcinoma, mesothelioma, neuroblastoma, lymphoma, or melanoma). Concerning the 189 samples with lung cancer, lung adenocarcinoma was diagnosed in 106 cases, followed by non-small cell lung cancer not-otherwise specified (NSCLC NOS) (44 cases), squamous cell carcinoma (20 cases) and neuroendocrine tumors (19 cases), including large cell neuroendocrine carcinoma (3 cases), small cell lung cancer (15 cases), and 1 case of carcinoid tumor (see Figure 1 and Table 1). In 11 cases, samples contained cells that were suspected to be malignant, but the immunocytochemistry did not confirm the malignancy, either because the sample was too necrotic or because the samples contained a low number of tumor cells (<1% of total cells). 

Among the 189 samples diagnosed with lung cancer, 72 (38.1%) were pleural effusions, 71 (37.5%) were lymph nodes collected by EBUS-TBNA, 29 (15.3%) were mediastinal or pulmonary masses collected by EBUS-TBNA, 6 (3.2%) were bronchoalveolar lavage fluids (BAL), 6 (3.2%) were pericardial effusions, 2 (1.1%) were cerebrospinal fluids (CSF), 2 (1.1%) were peritoneal effusion, and one (0.5%) was bone marrow. 

Patients were mostly diagnosed at stage IV (75.1%) and were current or former smokers (73%) (Table 1).

### 3.2. Molecular and PD-L1 Results

PD-L1 was performed on cytoblocks from 115 cytological samples. No results could be reported in 29 cases (25.2%) because there were less than 50 tumor cells. Of the 86 samples that could be analyzed, PD-L1 expression was found negative (<1%) in 22 cases (25.6%), between 1 and 49% in 34 cases (39.5%) and ≥50% in 30 cases (34.9%). Examples of these are shown in Figure 2. For 35 patients, the cytoblock was not performed because no sample remained after immunocytochemistry and NGS testing, or because the sample was too necrotic. For 20 patients, the cytoblock was available but PD-L1 was not performed on the cytoblock as it had already been done on a biopsy or on the resected tumor.

Next generation sequencing (NGS) was performed on 140 out of 150 cases (93%) diagnosed with lung adenocarcinoma or NSCLC NOS. *EGFR* mutation was found in 20 cases (14.2%), of which 12 received a tyrosine kinase inhibitor. The other 8 patients were not at stage IV, or were treated with the best supportive care, or the treatment was not known. *ALK* fusions were found in 4 (2.8%) cases and *ROS1* fusions in 2 (1.4%) cases. All these patients with *ALK* or *ROS1* fusions were treated with appropriate tyrosine kinase inhibitors. A mutation in *TP53* was found in 67 cases (47.8%), 46 of which had another associated mutation. *KRAS* mutation was found in 48 cases (34.3%), of which 29 had another associated mutation. Other mutations were found as *HER2*, *PIK3CA*, *STK11*, *RET*, *BRAF*, *DDR2*, *CTNNB1*, *SMAD4*, *PTEN* and *POLE* (Figure 3). The absence of mutation or fusion was found in 19 cases (13.6%). 

### 3.3. Improvement of Diagnosis and Therapeutic Management with Cytological Samples

We analyzed the impact of cytology results on the management of 189 patients with lung cancer: 

For 124 (65.6%) cases, the result was conducive to the diagnosis and allowed for therapeutic management. Among them, 37 cases had a biopsy and a cytological sample during the same procedure: the same diagnosis was obtained on biopsy and cytology for 24 patients, while for 13 patients, the biopsy was free of tumor cells and the diagnosis was only performed on cytology. In 3 cases, a biopsy was recommended after cytology because they were necrotic or because there was not enough material to perform the NGS. 

For 40 (21.2%) patients already known to have lung cancer, a cytological sample was performed in the context of suspected lung cancer progression (demonstrated by imaging). The presence of tumor cells in the cytological sample confirmed the progression and led to a change of therapeutic line.

For 19 (10.0%) cases, cytology results had no impact on the therapeutic management. In most of these cases, the metastatic site was already known (pleural, peritoneal, or pericardial) and the cytological analysis was performed because the effusion had to be drained. In other cases, the clinical status deteriorated rapidly, and the patient died within a few days. In 6 (3.2%) cases, no information was available. 

According to our data, the results obtained from cytological samples allow a therapeutic management for 87% of patients with lung cancer.

## 4. Discussion

In this study, we used cytospin specimens for staining and immunocytochemistry and cytoblocks for PD-L1 testing. The methods of cytological preparation each have their advantages and disadvantages. For example, cytoblocks can be compared to biopsies, while cytospin preparation provides good morphology that can easily be used for immunocytochemistry. Regardless of the method selected, rigorous quality controls are essential [6]. Studies report that small biopsies and cytological samples can account for up to 70% of specimens for the diagnosis of lung cancer [7]. In our study, immunocytochemistry on cytospin allowed to determine the type of cancer in 248 of the 259 cytological samples in which suspected tumor cells were observed. For the remaining 11 samples, the immunocytochemistry did not allow to characterize the cells either because of the quality (necrotic or too much altered cells), the lack of volume (e.g., cerebrospinal fluid), or the low number of suspected tumor cells (<1% of total cells). Despite this, the classification of tumor cells was successful in over 95% of cases. This rate shows the value of cytological samples for identifying tumor cells. Our results are consistent with other studies evaluating the ability of cytological samples to diagnose lung cancer [8,9,10]. For example, Rekhtman et al. [9] compared 192 pre-operative cytology specimens with histology and found a concordance of 96%. Proietti et al. [8] assessed the efficacy of lung cancer subtyping in cytology and biopsy samples from 941 patients and found a concordance in 92.8% of cases. Arnold et al. [11] conducted a prospective study to investigate the role of cytology in pleural effusions. They included 921 pleural effusions with 166 lung cancer and 100 lung adenocarcinomas. The sensitivity for the diagnosis of lung cancer was 56% and 82% for adenocarcinoma [11]. Others studies found similar results for the detection and the characterization of tumor cells in pleural effusion [12,13,14].

In the last decade, the emergence of immune checkpoint inhibitors and targeted therapies have modified the way in which cytological samples are managed. In addition to the diagnosis, the sample must allow for the assessment of PD-L1 expression and molecular testing. PD-L1 expression is a biomarker that predicts which patients are more likely to respond to immunotherapy. Immunotherapy can be prescribed in first line monotherapy for patients with advanced NSCLC and with ≥50% PD-L1 expression, and in second line therapy for metastatic NSCLC patients with ≥1% PD-L1 expression [15,16,17]. In this context, evaluation of PD-L1 expression is essential on cytological samples [18,19]. This test can be challenging; it needs an adequate protocol and quality controls [20]. For example, macrophages and mesothelial cells must be properly recognized to avoid counting them in the percentage of PD-L1 expressing cells. Cytoblocks are the most commonly used material for analysis and provide concordant results compared to biopsies, even if smears can also be used [21,22]. A recent multicenter study including 264 patients concluded that PD-L1 expression on cytological samples correctly predicts the efficacy of immunotherapy [23]. In our study, PD-L1 expression was tested only on 115 cytoblocks. When PD-L1 status has already been determined for a patient on biopsy or surgical specimen, the analysis was not performed again on cytological sample. 

Molecular testing must be performed for patients with advanced NSCLC as several oncogenic drivers are targetable. The International Association for the Study of Lung Cancer (IASLC) recommends to test *EGFR* mutations and *ALK* and *ROS1* fusions. *HER2*, *RET*, *MET*, *BRAF*, and *KRAS* are not indicated as a routine stand-alone assay but may be included in a large molecular testing panel [24,25]. In accordance, the European Society for Medical Oncology (ESMO) recommends the use of NGS that includes at least *EGFR* common mutations, *ALK* fusions, *MET* mutations, *BRAF* mutations, and *ROS1* fusions [24,26]. The absence of formalin in cytological samples facilitates molecular testing to be applied as NGS or polymerase chain reaction (PCR) [27]. Molecular testing on cytological samples has the advantage of providing results even if the sample volume or the number of tumor cells is low [28]. In our study, 93% of patients with lung adenocarcinoma or NSCLC had NGS results. Among them, only 13.5% did not show a molecular alteration in genes included in the tested panel. We have previously demonstrated the feasibility of detecting *ALK* and *ROS1* fusions from cytological samples either by immunocytochemistry completed by fluorescence in situ hybridization (FISH) if positive, or by NGS, with high concordance of both techniques [4,5]. *ALK* and *ROS1* fusions are now routinely performed by immunocytochemistry on cytological samples as part of diagnosis results [29]. Rekhtman et al. showed the feasibility of testing for *EGFR* and *KRAS* mutations in thoracic cytology [9]. In our study, the NGS is performed on the frozen cell pellets. The supernatant from post-centrifuge liquid based cytology can also be used for NGS [30]. 

The two main criteria that prevent all techniques (i.e immunocytochemistry, molecular testing and PD-L1) from being performed are low sample volume or the low representativeness of tumor cells. 

Regarding sample volume, pathologists should inform clinicians that the larger the volume sent to the laboratory, the more adequate the sample will be for diagnosis and additional testing. Particularly for effusions (pericardial, pleural, and peritoneal) where the puncture can evacuate up to several liters, the pathologist can receive only one or two milliliters. Adequacy of a standardized volume vary depending on the cellularity and the percentage of tumor cells. Currently, no standardized volume requirements exists, but several studies recommend 50 mL of fluid [31,32,33]. Dalvi et al. recommend at least 20 mL but demonstrated that the tumor cell proportion is critical for assessing diagnosis and molecular analysis [34]. The low number of tumor cells is a reason to perform a biopsy and obtain an accurate material for a new test. With a low number of tumor cells detected (1–5%), immunocytochemistry and NGS can potentially be interpreted [28]. But PD-L1 interpretation requires at least 100 tumor cells, otherwise pathologists are unable to obtain a result.

## 5. Conclusions

Over the last decade, major therapeutic advances in the treatment of lung cancer, with the introduction of targeted therapies and immune checkpoint inhibitors, have forced pathologists to change their practice and use cytological samples differently. Diagnosis alone is no longer enough and the pathologist must keep a portion of the sample to perform PD-L1 analysis and molecular testing. Cytological samples are obtained by minimally invasive procedures and can provide enough material for the diagnosis and the therapeutic management in patients with lung cancer. 

## Figures and Tables

**Figure 1 cells-12-00754-f001:**
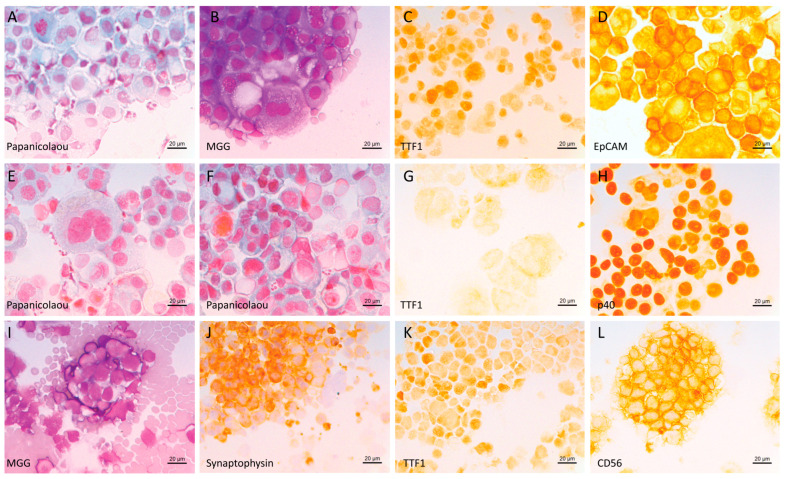
Example of lung cancer tumor cells (cytospin). (**A**–**D**) Cytology of pleural effusion from a patient diagnosed with lung adenocarcinoma: (**A**) Papanicolaou stain, (**B**) May-Grünwald–Giemsa (MGG) stain; (**C**,**D**) immunocytochemistry (ICC) (peroxidase staining) using antibodies against (**C**) TTF1 and (**D**) EpCAM. (**E**–**H**) Cytology of pleural effusion from a patient diagnosed with squamous cell carcinoma: (**E**,**F**) Papanicolaou stain; (**G**,**H**) ICC using antibodies against (**G**) TTF1 and (**H**) p40. (**I**–**L**) Cytology of a lymph node from a patient diagnosed with small cell lung cancer: (**I**) MGG stain, (**J**–**L**) ICC using antibodies against (**J**) synaptophysin, (**K**) TTF1 and (**L**) CD56. Black scale bar represents 20 µm.

**Figure 2 cells-12-00754-f002:**
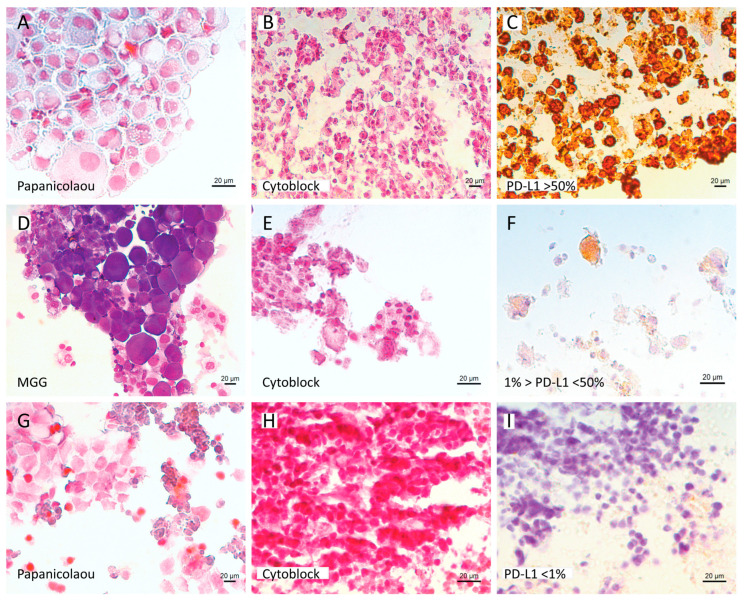
Example of PD-L1 expression results. (**A**–**F**) Cytology and PD-L1 expression results of pleural effusions from two patients diagnosed with lung adenocarcinoma (**A**–**C** patient 1, **D**–**F** patient 2). (**A**) Cytospin, Papanicolaou stain—Patient 1, (**B**,**E**) cytoblock H&E stain, (**D**) cytospin, MGG stain—Patient 2, (**C**,**F**) PD-L1 expression results (patient 1 (**C**) >50%, patient 2 (**F**) between 1 and 50%). (**G**–**I**) Cytology and PD-L1 results of a lymph node from patient diagnosed with NSCLC NOS. (**G**) Cytospin, Papanicolaou stain, (**H**) cytoblock H&E stain, (**I**) PD-L1 result (<1%). Black scale bar represents 20 µm.

**Figure 3 cells-12-00754-f003:**
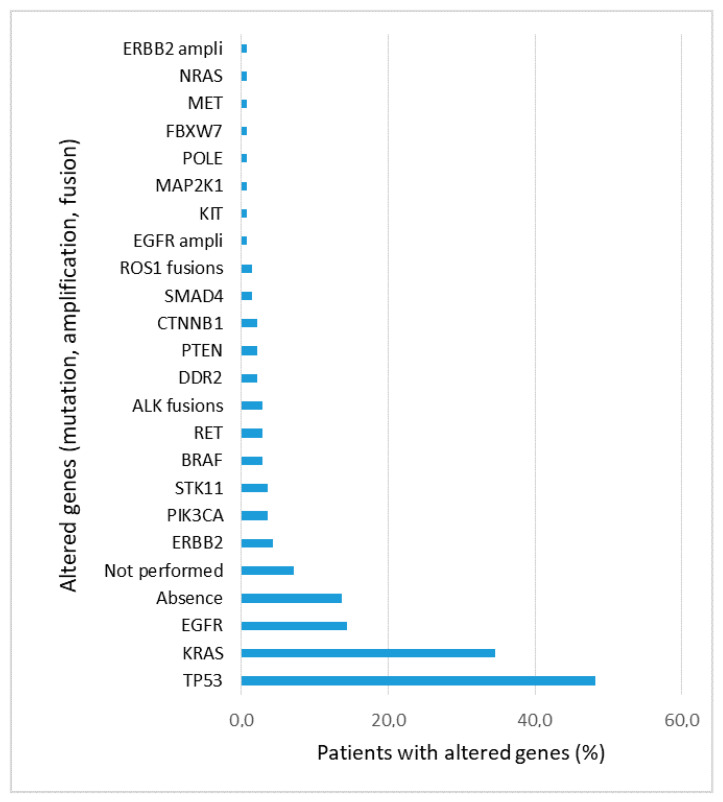
Percentage of patients with genetic alterations found in the 140 lung adenocarcinoma and NSCLC NOS samples and analyzed by NGS. For 10 patients, NGS was not performed.

**Table 1 cells-12-00754-t001:** Patients with lung tumor demographics.

	Parameter	*n* (%)
Age (years)	Mean ± SD	67 ± 11
	Range	32–91
Gender	Male	108 (57%)
	Female	81 (43%)
Histopathological type	Lung adenocarcinoma	106 (53%)
	Squamous cell carcinoma	20 (10%)
	NSCLC NOS	44 (22%)
	Small cell lung cancer	15 (7.5%)
	Large cell neuroendocrine carcinoma	3 (1.5%)
	Carcinoid tumor of the lung	1 (0.5%)
Smoking status	Never	32 (17%)
	Current/former	138 (73%)
	Unknown	19 (10%)
Stage	I	3 (1.6%)
	II	6 (3.2%)
	III	35 (18.5%)
	IV	142 (75.1%)
	Unknown	3 (1.6%)

## Data Availability

Not applicable.

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
