# Peer review of "Cytological Samples: An Asset for the Diagnosis and Therapeutic Management of Patients with Lung Cancer"

_cells, 2023, doi:10.3390/cells12050754_

Round 1

Reviewer 1 Report

This is an interesting article in which the authors demonstrate that Cytology samples can be used for the diagnosis and therapeutic management of patients with lung cancer. Although the diagnostic value of the Cytology sample is well-known, the application of PD-L1, a therapeutically important marker, in the cytology cell block with immunocytochemical staining is timely, commendable, and has practical clinical significance.   However, the overall quality of this paper can be improved by systematic grammatical and spelling editing.  A few examples are listed below:

1. On page 1, line 25, "Among this late" does not make sense.

2. On page 3, line 114, replace "i.e."  with "including" may be better.

3. On Page 6, line 170, it makes more sense to replace "than" with "as". 

4. On page 7, line 194, it makes more sense to replace "interest" with "value".

5. On page 7, line 216, "sand-alone" should be "stand-alone".

Numerous other small grammatical errors can be improved by professional editing. 

Reviewer 2 Report

The authors, based on a retrospective study, conclude that cytological samples obtained with minimally invasive procedure can provide enough materiel for the diagnosis and the therapeutic management of patient with lung cancer.

There are, however, several concerns, the main ones highlighted in bold.

Abstract

 I do not think it is correct that cytological samples are often the only source for diagnosis and determination of predictive markers. They are instead the easiest, more convenient and less invasive source.

Materials and methods

 Did the study focus on suspected lung cancers based on imaging or not? This is not clear and confusing that in a study finally focusing on lung cancer, 59/259 cases were not lung cancers.

 The criteria used to be included in this retrospective study should be detailed.

 PD-L1 evaluation is reported is in a separate paragraph than immunohistochemistry, while it's a also evaluated by immunohistochemistry. The paragraphs should be merged or named differently.

Results

 You write 200 samples with lung cancer while, out of the 259 samples, 59 were not lung cancers and cancer type was not determined for 11. Based on this, it should be 189 lung cancers and not 200, exactly as you reported it in the 3.3 subsection of the results section.

 When you detail the histological subtype for these 200 lung cancer samples, 106 + 44 + 20 + 19 = 189. So it supports my previous observation.

 However, when you describe the sampling technique for these 200 lung cancer samples, 76 + 75 + 30 + 7 + 6 + 3 + 2 + 1 = 200. So this very confusing. I guess you included here the 11 samples without a diagnosis. But all this has to be clarified.

 In Table 1, you should clarify what "suspicious" means (if I understood well, no possible diagnosis based on cytological samples).

 In Table 1, related to demographics of patients with lung tumors, you also have 200 patients, including the 11 "suspicious". Do you consider these 11 patients have lung cancer based on imaging or biopsies performed after the failure of cytology to get a diagnosis? This has to be clarified.

 Why only 74 samples analyzed PD-L1 expression? You provide an explanation in the discussion section but incomplete from my point of view.

 24% of these 74 samples were not evaluable for PD-L1, which appears high to me. This should be discussed in the discussion section.

 The percentage of negative PD-L1 expression is particularly low as compared to what has been published. Is there an explanation? This should be discussed in the discussion section.

 The scale bar is missing in Figure 2F.

 Why NGS analysis was not performed in 7% of samples? This should be mentioned and this proportion of samples should be added in Figure 3 to get the complete picture.

 In the sentence "Next generation sequencing (NGS) was performed on 140 out of 150 cases (93%) diagnosed with lung adenocarcinoma or non-small cell lung cancer (NSCLC)", it should be written NOS NSCLC.

 In the sentence "Among them, 37 had also a biopsy in the same time, 24 that give the same result than cytology and 13 that were negative (without tumor cells)", it is not clear to me which one was negative: cytology or biopsy?

 In the sentence "For 40 (21.2%) cases, the diagnosis was already known, but the cytological result led to a change of treatment", could you please mention why the cytological sampling was performed? Was it because the previous sample did not provide enough material for molecular analyses? If this was the reason, could you please report which sampling techniques were used (biopsy or cytology)?

 I'm not sure it is appropriate to include the samples obtained during palliative procedures and not for diagnosis or molecular analysis. Including these samples unfairly decreases the rate of cytological samples allowing therapeutic management to 87% of patients with lung cancer.

 That said, I'm surprised that the results are so good while there are so many samples for which PD-L1 expression has not been evaluated. 

 A title should be added to the supplementary tables. 

 It would have been nice to have a comparison, even if indirect because of the retrospective nature of the study, to results obtained with biopsies in the same institution.  

Discussion

 The comparison of the results of this study to those of previously published studies is poor.

 In the sentence "Our results are concordant with other studies showing the same range compared to histological results (92.8% to 100%)", it is not clear whether  these other studies evaluated cytological samples. 

 It is not correct to say that the first immunotherapy was pembrolizumab. It should be rephrased differently.

 Related to the sentence "In our study, PD-L1 expression was tested only on 74 cytoblocks because the test was not included in our routine protocol and was not performed if the result was already obtained on histological sample":

o Could you please report how many missing PD-L1 are related to the first reason and how many to the second one? In the results section, you mention 37 cases with biopsies performed before cytology.

o Could you please explain in which situations PD-L1 expression was not included in your routine protocol? Isn't it evaluated for all advanced-stage NSCLCs?

 In advanced-stage NSCLC, the guidelines for molecular testing are not correctly reported. 

General

 There are many grammatical errors.
